# On modelling the kinematics and evolutionary properties of pressure pulse driven impulsive solar jets

Balveer Singh, Kushagra Sharma, and Abhishek K. Srivastava

Department of Physics, Indian Institute of Technology (BHU), Varanasi-221005, India.

**Correspondence:** Balveer Singh (balveersingh.rs.phy17@itbhu.ac.in)

**Abstract.** In this paper, we describe the kinematical and evolutionary properties of the impulsive cool jets in the solar atmosphere using numerical simulation by Godunov-type PLUTO code at two different strength of the quiet-Sun magnetic field (B=56, 112 Gauss). These types of chromospheric jets are originated by the pressure pulse, which mimics after effects of the localized heating in the lower solar atmosphere. These jets may be responsible for the transport of mass and energy in the localized upper atmosphere (i.e., corona). The detection of the height-time profiles for the jets, originated by imposing the different pressure pulses, exhibit the asymmetric near parabolic behaviour. This infers that the upward motion of the jet occurs under the influence of pressure perturbation. However, its downward motion is not only governed by the gravitational free fall, but also due to the complex plasma motions near its base under the effect of counter propagating pulses. Maximum height and life-time of the jets w.r.t. the strength of the pressure pulse show a linear increasing trend. This suggests that if the extent of the heating and thus pressure perturbations will be longer then more longer chromospheric jets can be triggered from the same location in the chromosphere. For the certain amplitude of pressure pulse, the strong magnetic field configuration (B=112 Gauss) leads more longer jet compared to the weaker field (B=56 Gauss). This suggests that the strong magnetic field guides the pressure pulse driven jets more efficiency towards the higher corona. In conclusion, our model mimics the properties and evolution of variety of the cool impulsive jets in the chromosphere (e.g., macrospicules, network jets, isolated repeated cool jets, confined & small surges etc).

## 1 Introduction

Observations reveal that various types of plasma jets are ubiquitous in the solar atmosphere at diverse spatio-temporal scales (e.g., Sterling, 2000; Katsukawa et al., 2007; Shibata et al., 2007; Nisticò et al., 2009; Wedemeyer-Böhm et al., 2012; Tian et al., 2014; Raouafi et al., 2016; Kayshap et al., 2018, and references cited therein). These jets may play a significant role in transporting energy and mass in the localized upper atmosphere (i.e., corona) of the Sun. As far as the complex magnetic structuring of the solar chromosphere is concerned, it triggers various types of cool plasma jets (e.g., spicules, macrospicules, surges, magnetic swirls, network jets, evolution of plasma twists etc), which attribute to the association of exotic wave and

plasma processes (e.g., De Pontieu et al., 2004, 2007, 2014; McIntosh et al., 2011; Wedemeyer-Böhm et al., 2012; Tian et al., 2014; Kayshap et al., 2018; Srivastava et al., 2017, 2018, and references cited therein).

In the recent observations, it is found that quiet solar chromosphere triggers various kinds of localized solar jets (e.g., magnetic swirls, network jets, spicule-like rotating plasma structures, etc) apart from the typical spicules (e.g., Wedemeyer-Böhm et al., 2012; Tian et al., 2014; Shetye et al., 2016, and references cited therein). Moreover, solar macrospicules, confined surges, anemone jets etc are also some other flowing magnetic structures typically observed in the solar chromosphere (e.g., Roy, 1973; Wilhelm, 2000; Morita et al., 2010, and references cited therein). The evolution of such a variety of chromospheric jets at diverse spatio-temporal scales provides a very detailed picture of the ongoing different plasma processes triggering these jets in an abundant measure in the localized solar atmosphere (e.g., De Pontieu et al., 2004, 2007; McIntosh et al., 2011; Kayshap et al., 2013b; Murawski et al., 2011, 2018; Srivastava et al., 2017, 2018; Martínez-Sykora et al., 2017; Iijima and Yokoyama, 2017; Kayshap et al., 2018; Liu et al., 2019, and references cited therein).

It is well established that complex magnetic structuring of the solar chromosphere enables small-scale magnetic reconnection and thus the localized heating and generation of the solar jets (e.g., Yokoyama and Shibata, 1995). However, the height of the reconnection and its magnitude decides the evolution of the inherent physical processes. If the reconnection occurs in the chromosphere, then the evolution of the shocks and plasma motions may be evident (e.g., Shibata et al., 1992; Kayshap et al., 2013a, and references cited therein). However, the reconnection height lying in the inner corona may result in the jet propagation due to the direct influence of Lorentz force, and Alfvén waves can also be evolved (e.g., Nishizuka et al., 2008; Jelínek et al., 2015, and references cited therein).

Recently, observations and modeling of the chromosphere revealed that many localized jets (e.g., spicules and macrospicules, network jets, confined surges, jets in the twisted or straight magnetic spires, chromospheric jets near flare ribbons etc) exhibit the brightening and heating at their footpoints before their evolution (e.g., Isobe et al., 2007; Murawski et al., 2011; Uddin et al., 2012; Kayshap et al., 2013b, 2018; Li et al., 2019, and references cited therein). There may also be the shock related brightening in the lower solar atmosphere at the site of the evolution of such jets. Martínez-Sykora et al. (2011) and Judge et al. (2012) have studied the structure and evolution of the spicule-like jets that are triggered by the pulses which are launched in the vertical velocity component from the upper chromosphere. The vertical velocity pulse is converted into a shock that propagates in the upward direction and chromospheric plasma follows these shocks to form the jet (e.g., Martínez-Sykora et al., 2009; McIntosh, 2007, and references cited therein). These shock-driven jets show a thin linear plasma structure that reaches a few Mm above the chromosphere and can transport mass and energy in the overlying solar atmosphere. Kuźma et al. (2017) have also proposed the two fluid spicule model. These spicules are triggered by the velocity pulse in vertical direction, while radiative cooling and thermal conduction do not play any significant role in the dynamics of such plasma ejecta. Recent observations reveal that the network jets are found to be associated with the impulsive origin in the chromosphere (e.g., Kayshap et al., 2018), and they morphologically overlap with each other and with type-I/type-II spicules as well in the quiet Sun chromosphere (e.g., Tian et al., 2014). In the present paper, we present a model of the pressure-pulse driven jets and their evolutionary properties, which may mimic the variety of impulsive chromospheric jets (e.g., macro spicules, network jets, confined smaller jet-like surges of moderate length etc) originating at the top of the photosphere. The pressure pulse acts as a

driver mimicking that heating is already occurred locally and activated the pressure perturbations launching these jets. We have also estimated typical parameters (height, width, lifetime) of the jets triggered by different pressure pulses. The kinematics and evolutionary properties of such jets have also been estimated for two magnetic field strengths of the quiet-Sun and comparison has been made. In Sect. 2, we describe the MHD model of the jet, equilibrium model of the solar atmosphere, perturbations, and numerical methods. The results are described in Sect. 3. The discussion and conclusions is outlined in the last section.

## 2 MHD Model of the Pressure Pulse Driven Jets

In order to model the pressure pulse driven solar jets in the quiet-Sun, we consider and implement a gravitationally-stratified and magnetized solar atmosphere. This atmosphere is approximated by the ideal MHD equations in their conservative form as outlined below (e.g., Mignone et al., 2007, 2012; Wołoszkiewicz et al., 2014):

$$\frac{\partial}{\partial t}\begin{pmatrix} \rho \\ \rho\mathbf{v} \\ E \\ \mathbf{B} \end{pmatrix} + \nabla \cdot \begin{pmatrix} \rho\mathbf{v} \\ \rho\mathbf{v}\mathbf{v} - \frac{\mathbf{B}\mathbf{B}}{\mu} + \mathbf{I}p_t \\ (E+p_t)\mathbf{v} - \frac{\mathbf{B}}{\mu}(\mathbf{v}.\mathbf{B}) \\ \mathbf{v}\mathbf{B} - \mathbf{B}\mathbf{v} \end{pmatrix} = \begin{pmatrix} 0 \\ \rho\mathbf{g} \\ \rho\mathbf{v}.\mathbf{g} \\ 0 \end{pmatrix} \tag{1}$$

In the above representation of the set of ideal MHD equations, the symbol $\rho$ depicts mass density in the solar atmosphere, $\mathbf{v}$ denotes the velocity, $\mathbf{B}$ is the magnetic-field satisfying $\nabla.\mathbf{B} = 0$ divergence free condition, and $\mu$ depicts the magnetic permeability. The parameter $p_t = p + \mathbf{B}^2/(2\mu)$ defines the total pressure which is an addition of thermal (p) and magnetic ($\mathbf{B}^2/2\mu$) pressure. $\mathbf{I}$ is $3 \times 3$ unit matrix. The quantity E describes the total energy, which is given as follows in the Eq. (2) (e.g., Mignone et al., 2007, 2012; Wołoszkiewicz et al., 2014).

$$E = \frac{p}{\gamma - 1} + \frac{\rho\mathbf{v}^2}{2} + \frac{\mathbf{B}^2}{2\mu} \tag{2}$$

In this equation, $\gamma$ is the specific heats ratio. PLUTO code also satisfies an ideal gas equation under the MHD approximation.

$$p = \frac{k_B}{m}\rho T \tag{3}$$

In this Eq. (3), T symbolizes temperature while $k_B$ is the Boltzmann constant. The symbol 'm' denotes the mean particle mass. We take typical value of 'm' for the model atmosphere that is equal to 1.24 . We do not consider the non-ideal conditions such as velocity of the background plasma flows, dissipative effects like viscosity and resistivity, magnetic diffusivity, cooling and/or heating of the plasma, because we are interested in understanding the kinematics and evolutionary properties of these pressure driven jets. We consider $\nabla \cdot \mathbf{B}$ equals to zero or a very negligible values of it in the numerical domain.

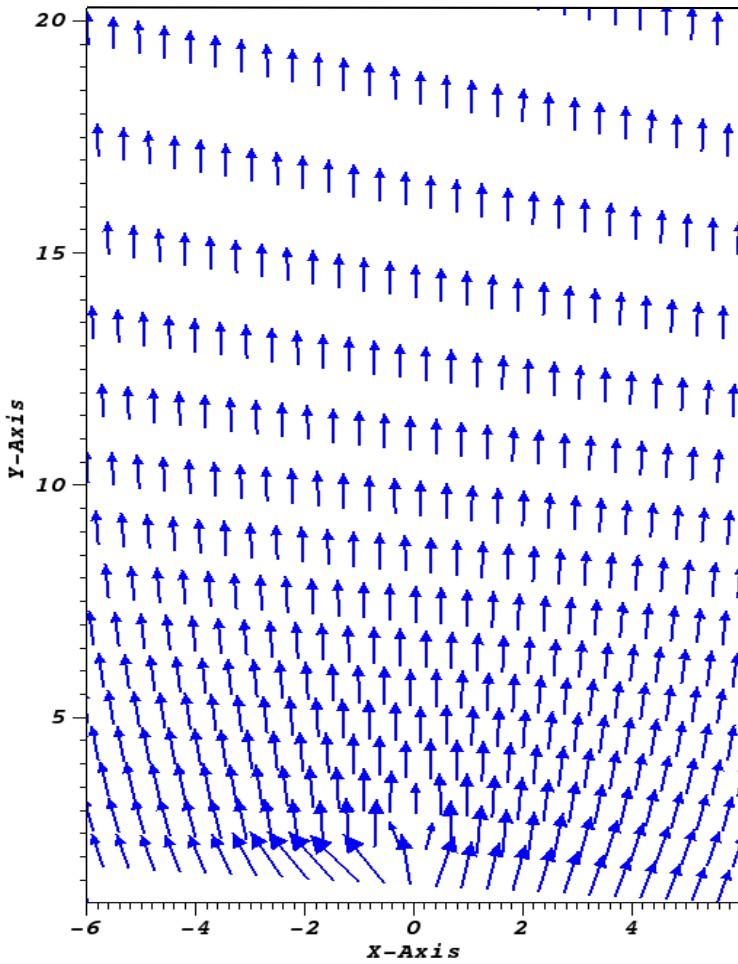

**Figure 1.** Equilibrium open and expanding magnetic field lines in the model quiet-Sun solar atmosphere. X-axis and Y-axis are given in Mm.

## 2.1 Equilibrium Model of the Solar Atmosphere

We consider the numerical simulation box mimicking a realistic solar atmosphere covering the region from the photosphere to the inner corona. In the case of the initial solar atmosphere subjected to the hydrostatic equilibrium, the realistic temperature profile $T_e(\mathrm{y})$ is shown in Fig. 2, which is **inferred from the observed line profiles by Avrett and Loeser (2008) and also depicted in Konkol et al. (2012) and Wołoszkiewicz et al. (2014)**.

In the case of the static equilibrium of the solar atmosphere, i.e., when there is no background initial flows ($V_e = 0$), it is approximated as a force free as well as a current free magnetic field. Therefore, we can abbreviate the initial magnetic field condition as

$$\nabla \times \mathbf{B}_e = 0, \qquad\qquad (\nabla \times \mathbf{B}_e) \times \mathbf{B}_e = 0 \qquad\qquad (4)$$

In the given expression, the subscript $'e'$ represents the equilibrium quantities. The vector magnetic potential can be estimated with the magnetic flux function given as (Konkol et al., 2012; Wołoszkiewicz et al., 2014):

$$\mathbf{B}_e(x,y) = [\mathbf{B}_{ex}, \mathbf{B}_{ey}, 0] = \nabla \times (A_e \widehat{z}) \qquad\qquad (5)$$

Where, the magnetic flux function is given as follows,

$$A_e(x,y) = \frac{(x-a)B_{ref}}{(x-a)^2 + (y-b)^2} \qquad\qquad (6)$$

Here $B_{ref}$ is the strength of the pole, and $a$ and $b$ collectively determine its position. In the above mentioned expression, we fix the vertical coordinate of magnetic pole, b = -5 Mm in the convection zone. The jet is triggering from the chromosphere (1.8 Mm) along the open magnetic field lines and the pulse is initially lunched there. Thus, there is no dynamics occurred in the convection zone related to the evolution of these isolated chromospheric jets. Our realistic temperature model (Fig. 2) smoothly extends from the convection zone to the photosphere and thereafter it couples to the chromosphere, TR, and corona.

In order to save the computing time, we do set lower boundary of the simulation box at the photosphere. For the visualization, we have already given the magnetic field variations in the plot (see Fig. 1 for B=112 Gauss pole strength) starting from the phtosphere to the corona. It clearly shows that the magnetic pole is set somewhere in the convection zone deeper, and the magnetic field is smoothly extended into the solar atmosphere at higher heights with an exponential decay. Putting the pole deep below in the convection zone is the requirement, of the numerical simulation as we keep the magnetic singularity away

from the simulation box in order to set an appropriate initial force and current free magnetic atmosphere. The quiet-Sun cool chromospheric jets are simulated in our model. We set the source magnetic field of the typical order in the quiet-Sun. Moreover, the chosen magnetic fields smoothly extend to the inner corona and appropriately set the reasonable values of plasma beta and $Alfvén$ speed (Fig. 3, 5). This helps in the evolution of the perturbations and the launch of the collimated jets. In this model, we have chosen two different configurations of magnetic field strength, i.e., B=56 Gauss and B=112 Gauss. The reference

level is taken in the overlying corona at $y_{ref}$ = 10 Mm. The magnetic field lines exhibit open and expanding field behaviour as shown in Fig. 1. We take the magnetic field strength typical of the regions at the Sun where chromospheric jets are formed in the quiet-Sun.

In the hydrostatic equilibrium of the solar atmosphere set in the simulation box, the gravity force balances the pressure gradient force. This can be written as follows:

$$-\bigtriangledown p + \rho \mathbf{g} = 0 \qquad\qquad (7)$$

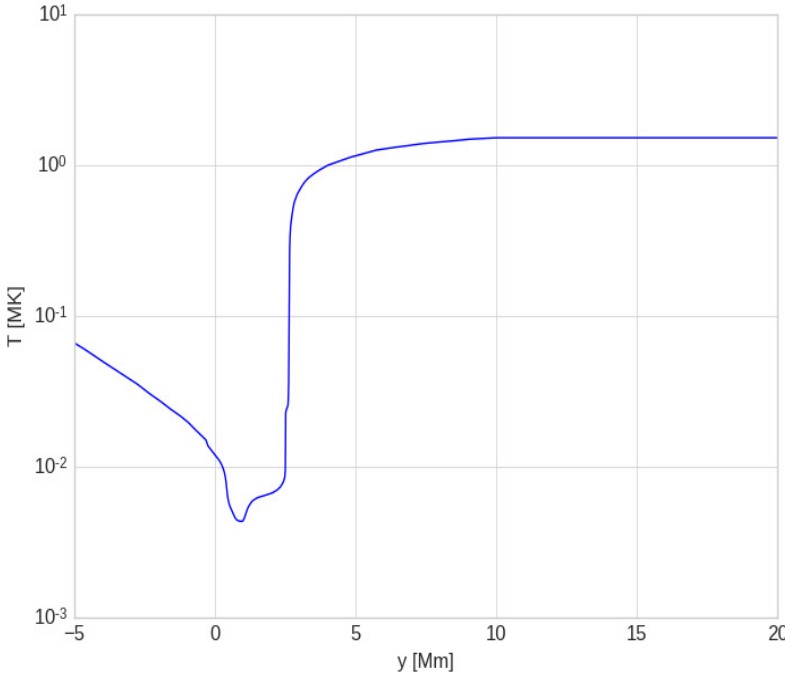

**Figure 2.** Temperature variation with the height in the initial static solar atmosphere.

Here we keep fix the value of **g** as 27400.0 cm $s^{-2}$. Using the vertical component of the hydrostatic equilibrium in the model solar atmosphere and ideal gas law, we determine the equilibrium plasma gas pressure and mass density as follows (Konkol et al., 2012; Wołoszkiewicz et al., 2014):

$$p_e(y) = p_{ref} exp \left( - \int_{yref}^{y} \frac{dy'}{\Lambda(y')} \right), \qquad \rho_e(y) = \frac{p_e(y)}{g\Lambda(y)} \qquad (8)$$

where

$$\Lambda(y) = \frac{k_B T_e(y)}{\hat{m}g} \qquad (9)$$

In these expressions, $P_{ref}$ is the gas pressure at reference level $y_{ref}$, which is attributed as the pressure scale height.

For considering the gravitationally stratified and longitudinally structured solar atmosphere, we obtain the plasma temperature profile $T_e(y)$ which is derived by Avrett and Loeser (2008) and displayed in Fig. 2. It should be noted that the typical value of the temperature $T_e(y)$ at the top of the photosphere is about 5700 K. This temperature corresponds to y = 0.5 Mm at the photosphere, while it falls gradually and attains its minimum about 4350 K at higher altitudes about y = 0.95 Mm which

represents a temperature minimum. As we move higher up in the solar atmosphere, $T_e(y)$ increases gradually with the height up to the transition region which is located at $y \simeq 2.7 Mm$. $T_e(y)$ sharply increases up to the corona and finally attains the constant value of mega-Kelvin at the coronal heights as shown in Fig. 2.

The magnetic field and plasma-beta at y = 2.7 Mm (at the transition region) are 6.884 Gauss and 0.0005 respectively while they grow with the depth and attains 7.646 Gauss and 0.219 respectively at y = 1.5 Mm, which is located within the chromosphere (Fig. 3).

Using Eq. (8), we obtain the equilibrium mass density and gas pressure profiles w.r.t. height (Fig. 4). We found that the equilibrium mass density and gas pressure at 2.7 Mm (at the transition region) are $4.595 \times 10^{-12}$ kg $m^{-3}$ and $1.26 \times 10^{-3}$ Pascal respectively. They grow with the depth and attain $1.537 \times 10^{-7}$ kg $m^{-3}$ and $6.411 \times 10^{-1}$ Pascal respectively at y = 1.5 Mm which is located within the chromosphere. Fig. 5 shows the trend of $Alfvén$ speed and sound speed w.r.t. height in the solar atmosphere. The $Alfvén$ speed is calculated by $V_A = \sqrt{\frac{B^2}{\mu \rho}}$, which is 1015.583 km $sec^{-1}$ at y = 2.7 Mm (i.e., transition region) while it decrease with the depth and attains 6.167 km $sec^{-1}$ at y = 1.5 Mm located within the chromosphere. The sound speed calculated by $V_s = \sqrt{\frac{\gamma p}{\rho}}$ at the transition region i.e. 2.7 Mm is 21.392 km $sec^{-1}$ while it decreases with the depth and attains 2.638 km $sec^{-1}$ within the chromosphere at y = 1.5 Mm.

These background physical quantities (i.e., density, pressure, magnetic field plasma-beta, $Alfvén$ speed and sound speed) for the model gravitationally-stratified and magnetized solar atmosphere are appropriately set in the model solar atmosphere. We have shown these profiles for the atmosphere which have source magnetic field value of 112 Gauss. The various physical quantities, their values, and variations with the height (y) in the structured solar atmosphere clearly indicate their smooth extension into the inner corona. Their reasonable values are set for the model atmosphere, and are appropriate for launching the perturbations and associated jets.

### 2.1.1 Perturbations

We consider the initial solar atmosphere as in the hydrostatic equilibrium and gravitationally-stratified. We perturb the equilibrium atmosphere by the initial pulse in the vertical direction in equilibrium gas pressure. The Gaussian form of the pressure pulse in the vertical direction is given as follows:

$$P = P_0 \left[ 1 + A_p \times exp \left( -\frac{(x - x_0)^2 + (y - y_0)^2}{w^2} \right) \right], \tag{10}$$

Here $(x_0, y_0)$ is the initial position of the pressure pulse. w is the width of the pulse, and $A_p$ denotes the pressure amplitude. We fix the value of $x_0, y_0$ as 0 and 1.8 Mm respectively, therefore, launching the pressure perturbations in the chromosphere. We take w equals to 0.2 Mm. We imply different pressure pulse strengths $A_p$ = 4-22, which account for the generation of the variety of solar jets.

Observations reveal that these types of chromospheric jets are found to be associated with the impulsive origin in the chromosphere, and their footpoints are usually associated with the brightening. The chosen pressure pulses in this modeling mimic the after effect of the heating scenario at their footpoints before evolution. This is not a direct implementation of the heating as we

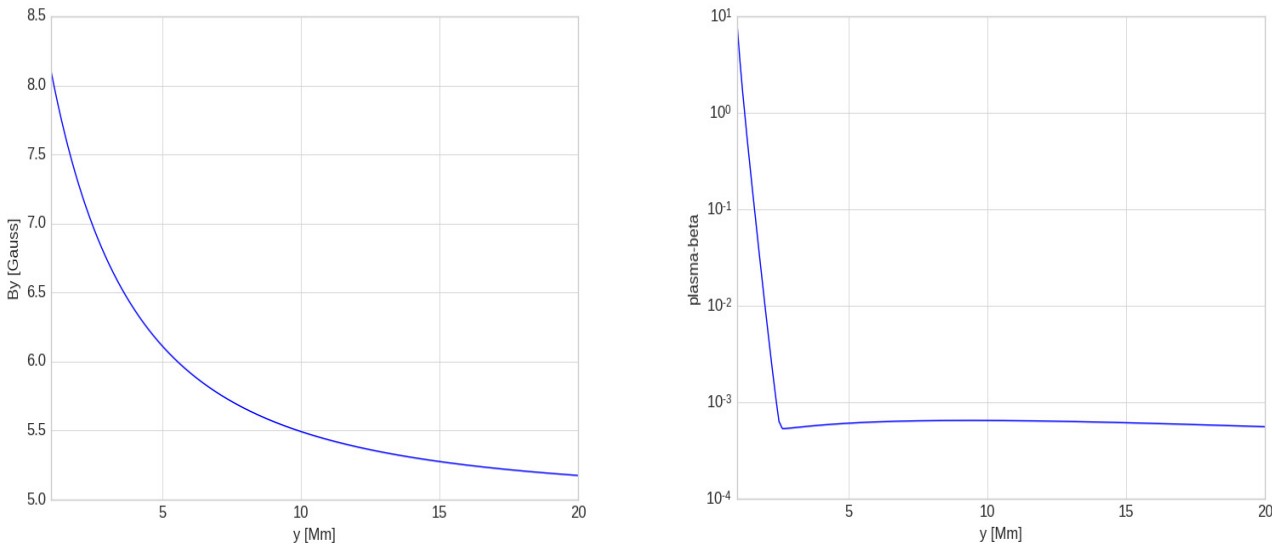

**Figure 3.** Magnetic field profile (left) and plasma-beta profile (right) vs. height (y) in the model quiet solar atmosphere.

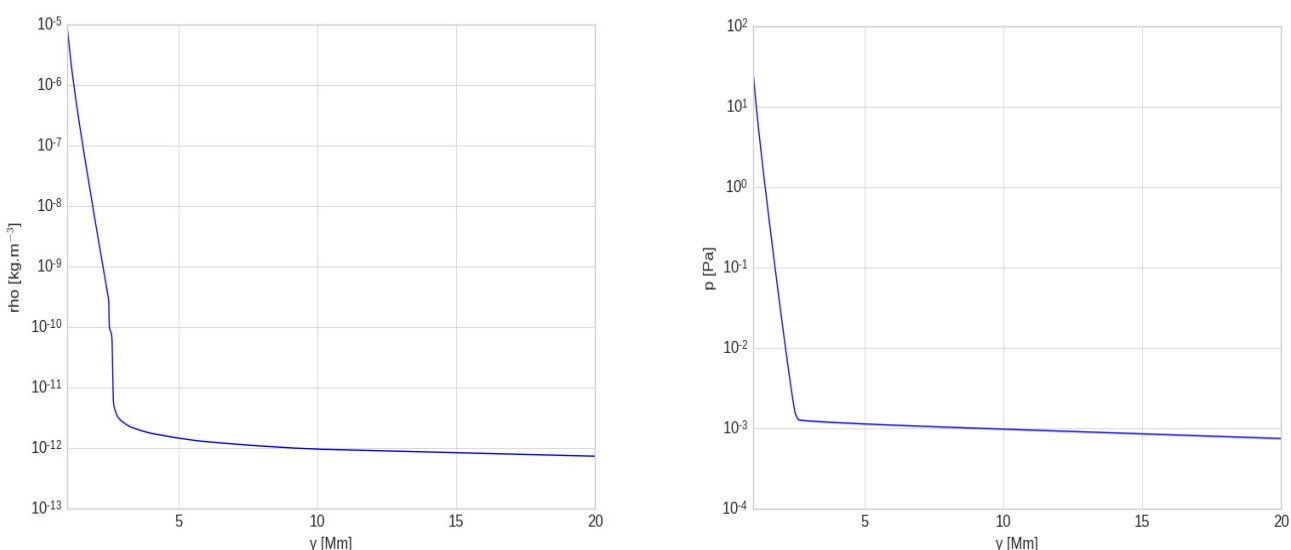

**Figure 4.** Equilibrium profiles of mass density (left) and gas pressure (right) vs. height in the model quiet solar atmosphere.

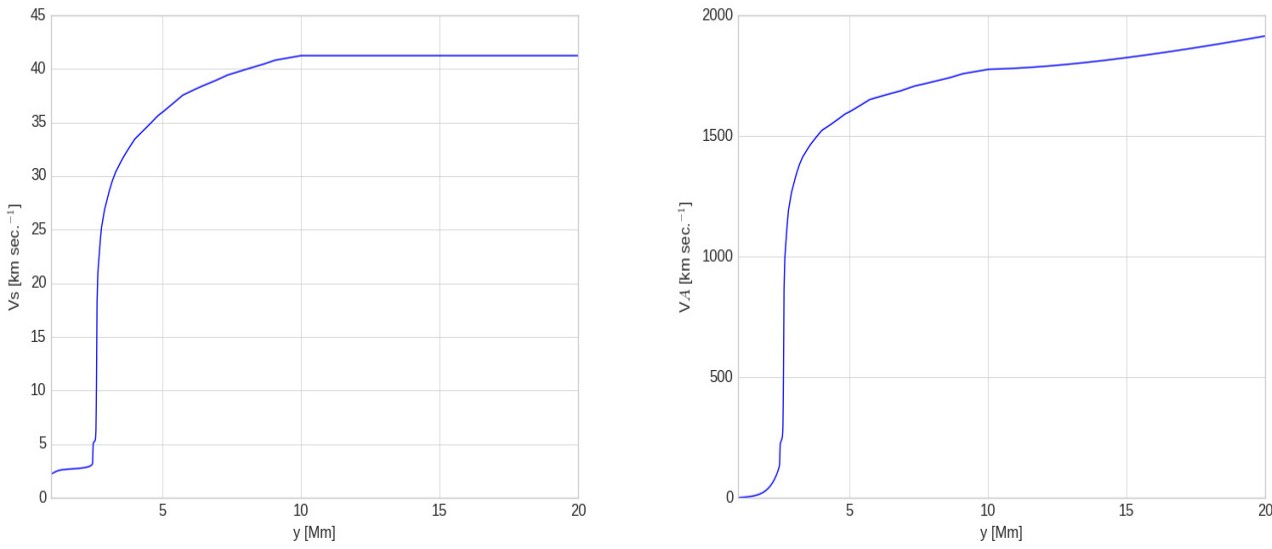

**Figure 5.** Sound speed (left) and $Alfvén$ speed (right) vs. height (y) in the model quiet solar atmosphere.

use the ideal set of MHD equations in the numerical simulation. However, the Gaussian pressure pulse implies the disturbance that represents the response in the form of the pressure perturbations once the underlying heating episode is completed. This perturbation locally alters the equilibrium and generates the propagating disturbances. The pressure pulse is chosen with the varying amplitude $A_p$ = 4-22 to understand the evolutionary and propagation properties of the cool chromospheric jets. We perform the numerical simulation at two strength of magnetic fields (i.e., B=56, 112 Gauss), typical of quiet-Sun atmosphere.

### 2.2 Numerical Methods

PLUTO code is a Godunov-type, non -linear, **finite-volume / finite-difference** code which takes into account ideal and non-ideal sets of governing equations (Mignone et al., 2007, 2012; Wołoszkiewicz et al., 2014) . It is constructed to integrate numerically a system of conservation laws, which can be shown as follows:

$$\frac{\partial \mathbf{U}}{\partial t} + \nabla \cdot \mathbf{F(U)} = \mathbf{S(U)} \tag{11}$$

In this equation, **U** denotes a set of conservative physical fields (e.g., magnetic field, density, velocity, pressure etc), while **F(U)** is the flux tensor and **S(U)** is the source term.

PLUTO code considers to utilize a second-order unsplite Godunov solver and Adaptive Mesh Refinement (AMR) of the system of conservation laws as shown in the Eq.(11). In order to solve the set of ideal MHD equations (Eq. 1) numerically, we set the simulation box as (-6, 6) Mm × (1, 21) Mm. This represents a realistic localized solar atmosphere of 12 Mm and 20 Mm span respectively in the horizontal and vertical directions. This solar atmosphere is constructed within the simulation box, and all four boundary conditions by fixing the simulation region to their equilibrium values. Numerical simulation is done with

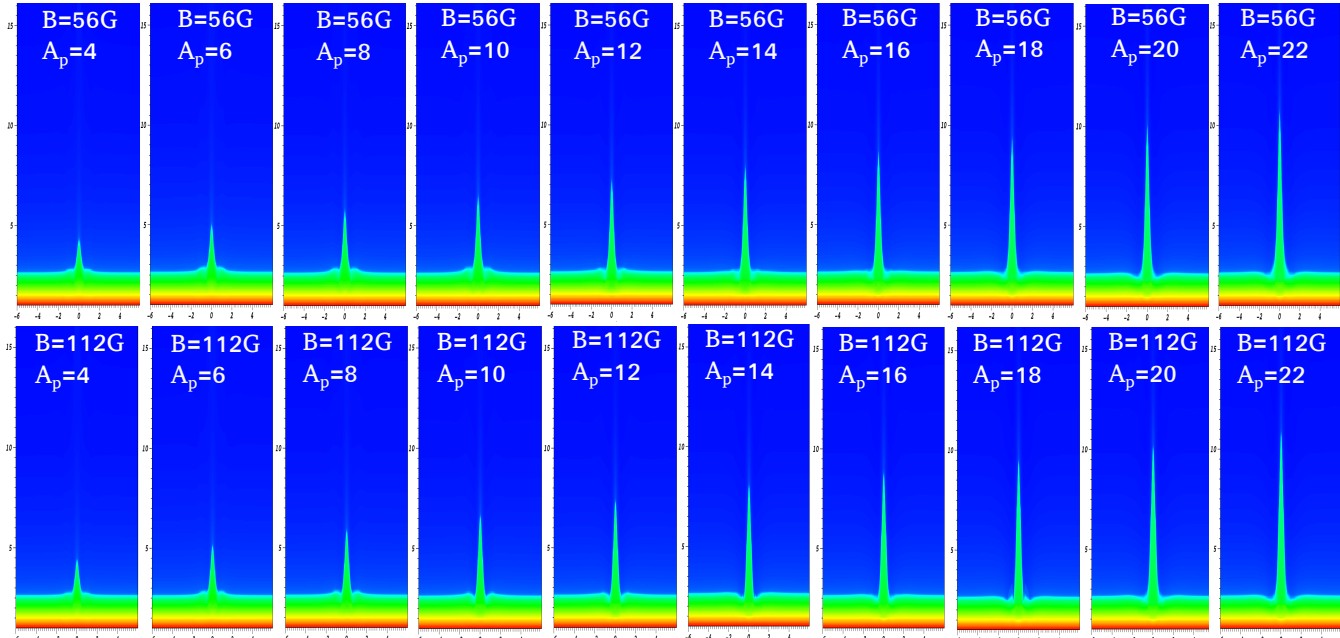

**Figure 6.** Evolution of the plasma jets at different pressure pulse in two different strength of the magnetic fields B=56 Gauss and B=112 Gauss. The Mosaic diagram shows the maximum height of the evolved jets at different pressure pulses, e.g., $A_p$ = 4-22. Horizontal (x) and vertical (y) axes are in Mm.

double precision arithmetic using multiple passage interface (MPI) (Mignone et al., 2007).The eight processors are used in the parallel calculations. It took approximately 12 hours of CPU time for each set of the calculations. We adopted a static uniform grid which is divided into 384 equal cells in x-direction. we also adopted a static uniform and stretched grid which are divided into 128 equal and 256 equal cells respectively in y-direction. The resolution of the simulation domain is 31 km per numerical cell. We have obtained the numerical simulation data every 10 seconds.

In our modelling, we set the Courant-Friedrichs-Lewy (CFL) number equals to 0.25. We use Roe solver for the flux computation, which is lineraized Riemann solver based on the characteristic decomposition of the Roe matrix (cf. Mignone et al., 2007, 2012).

## 3  Results

The pressure perturbations in the solar chromosphere launch the thin and localized plasma jets (Fig. 6). These jets carry the mass from the chromosphere to the outer corona along the expanding open field lines. The pressure perturbations mimic the impulsive origin of these jets due to the localized heating episodes that might occur at some time epoch in the solar chromosphere and that cause the direct pressure disturbance in the equilibrium atmosphere. As the pressure pulse is launched, the plasma gets essential velocity perturbations guided along the vertical and expanding field lines. These perturbations are converted in

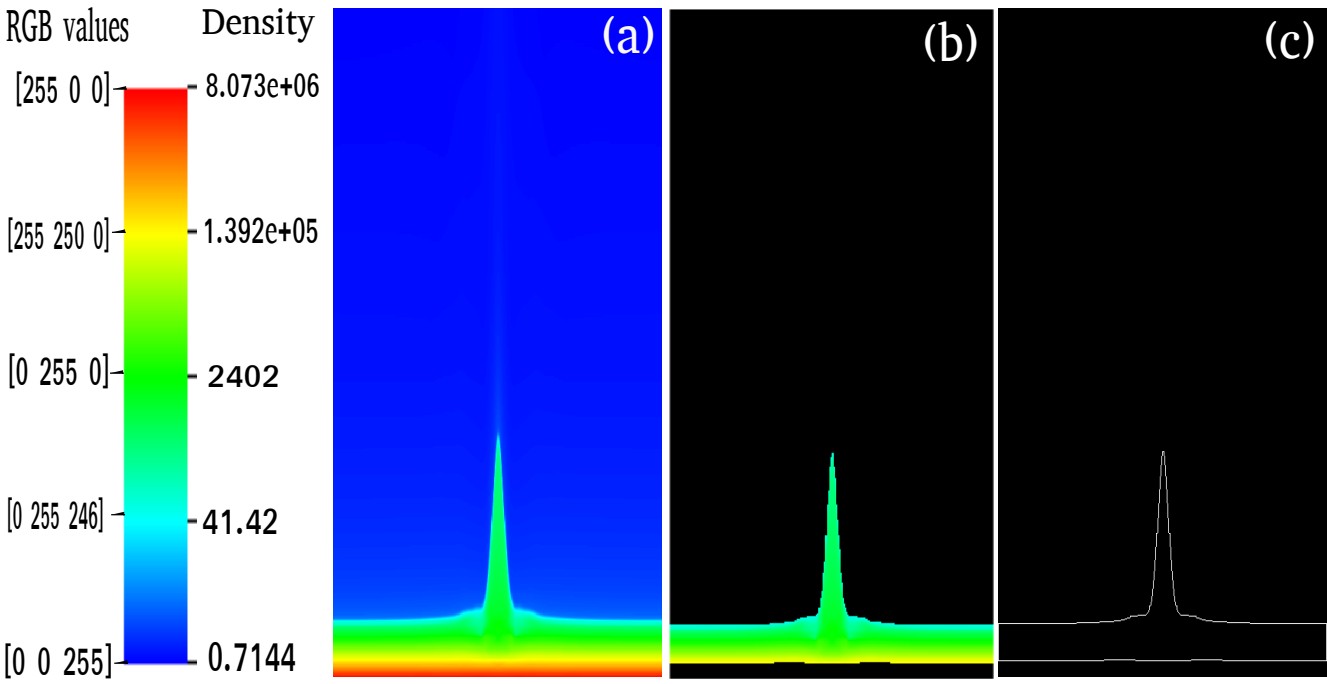

**Figure 7.** Automated detection of the plasma jet in the numerical simulation data to establish its time-distance profile and termination point.

magneto acoustic shocks in the stratified atmosphere which is followed by the motion of cool plasma behind. Shock leaves the domain quickly, and resulting effect that remains behind is the formation of the jet. The pressure pulse driven cool jets depend on the amplitude of the initial pulse ($A_p$) as well as the height of its triggering. This further converts into the acoustic shocks that trigger plasma into the upward direction. The low-pressure region is created behind the slow magnetoacoustic shock, which

drives the cool plasma from the chromosphere upward mimicking cool jet. Therefore, the amplitude of the pressure pulse as well as its triggering site are important factors upon which the evolution of the jets depends. In this model, we have studied chromospheric cool jets, so we can take a certain amplitude of the pressure pulse at the chromospheric height below the TR to generate these plasma ejecta.

We have analyzed the kinematical and evolutionary properties of these pressure pulse driven solar jets by their detection

(Fig. 7). Fig. 8 displays the height-time profile of various jets at two different configurations of the magnetic field (B = 56 Gauss and B = 112 Gauss). It shows that the jets exhibit asymmetric parabolic paths, which is different than the normal parabolic path of any ejecta in the gravitational field. We develop a code in Matlab to find the height of the single isolated jet with time and its termination point. For calculating the height of the jet, we took the time-series of the numerical simulation data (Fig. 7a). In this density map, we estimate the assigned RGB values corresponding to the density variations over the colour

scales. Highest density corresponds to the fixed value of the green with G=255 and the value of it decreases towards blue (B) as we go towards the tip of the jet with decreasing density. Ultimately it becomes 245 as the jet ends. Although it fades away slowly and there is no sharp boundary, so we take B=245 as the approximate boundary value for the jet. We have used this

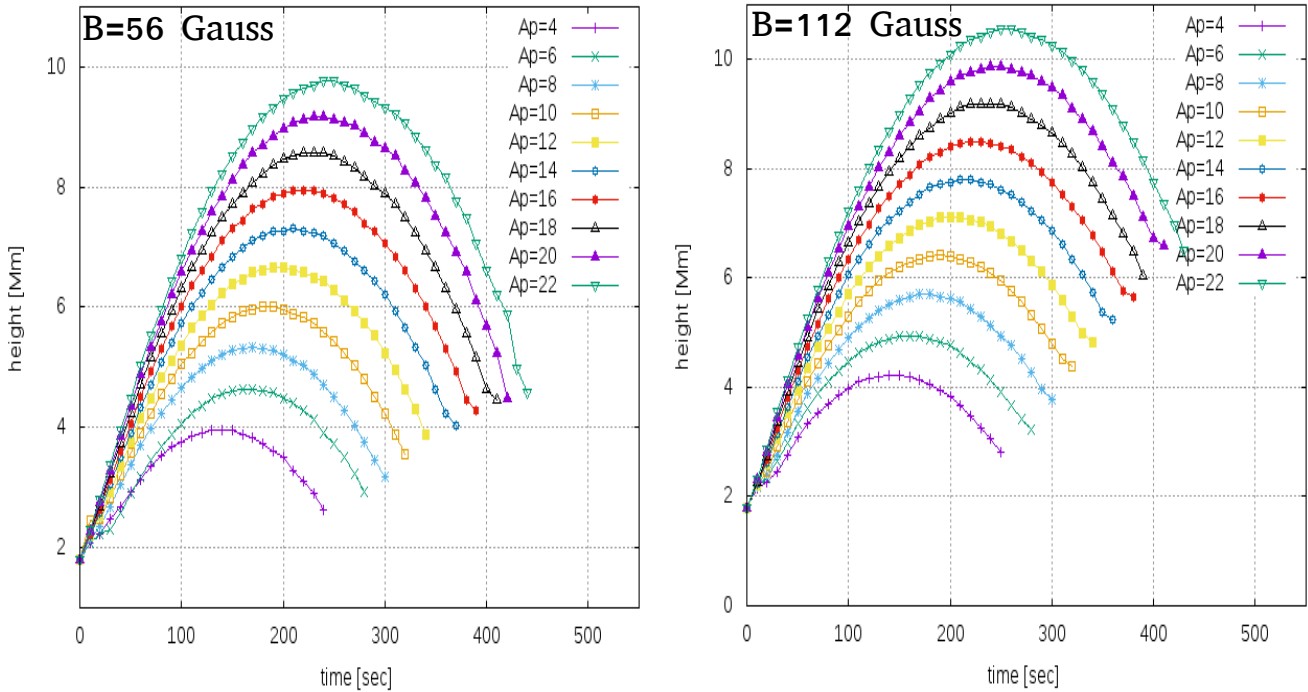

**Figure 8.** Height of the jets and their evolution w.r.t. the time.

observation to calculate the required parameters by fixing G=255, B<= 245 and generated a masked image with only G=255, B<=245 and rest of the portion is left as black (Fig. 7b). Then, we have generated the contour of the masked image to get Fig. 7c, and thereafter we found the height variations automatically w.r.t. time. We took the snapshots of density map for the calculation of height when the jet has reached above the transition region, and the snapshots of pressure map for the calculation of the height of the jet when it was below the transition region. Also at time t=0 s, we took the value of height as 1.8 Mm as was set in the simulation. The lifetime of the isolated jet is estimated in the masked time-series where only the jet's motion is visible. We calculate the time when the height of the jet again starts increasing after the fall, and this point we consider as the termination point of the first isolated jet. For $A_p$=14 (time=190 sec and corresponding density map), we have demonstrated the above mentioned procedure in Fig. 7, to detect the jet.

We investigate the evolution of the pulses and associated jets at two different strength of the magnetic field (B = 56, 112 Gauss). The automatically detected height-time profiles for the jets originated by imposing different pressure pulses clearly exhibit the asymmetric near parabolic profiles (Fig. 8). This infers that the upward motion of the jet under the influence of pressure perturbations does occur, however, its downward motion is not singly governed by the gravitational free fall. The downward propagating counterpart of the perturbation when reflects from the photosphere, it goes up and causes its interaction with the falling jet plasma. This creates the complex scenario near the base of the jet where we observe the complex plasma motion and evolution of the multiple jets one by one (e.g., Murawski et al., 2011; Kayshap et al., 2018; Li et al., 2019). We aim

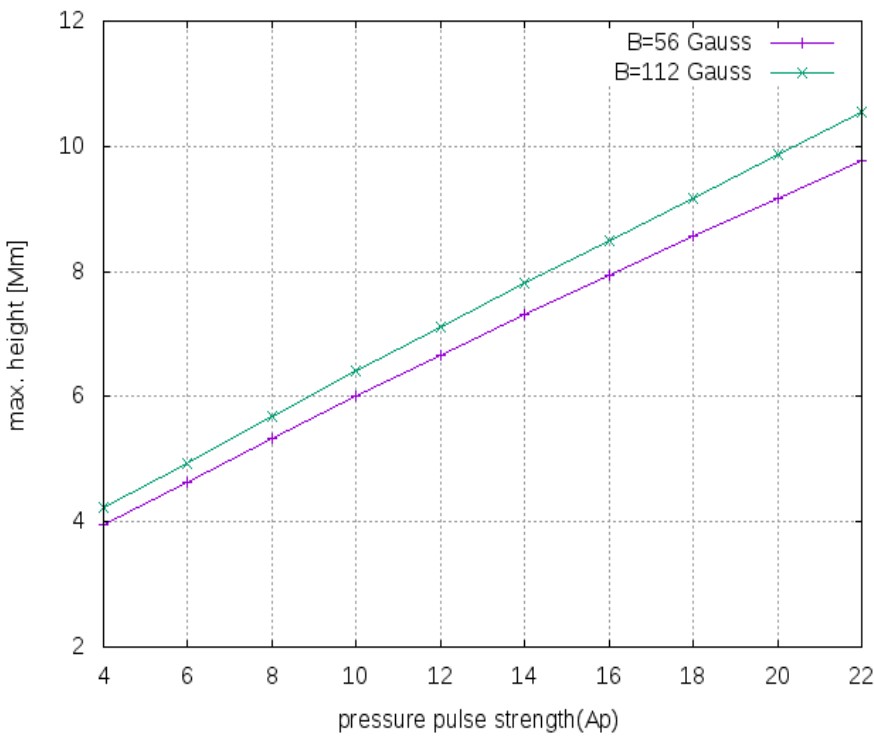

**Figure 9.** Maximum height of the model jets vs. pressure pulse strength.

to study the evolution of the first isolated jet that is generated due to the forward propagating component of the disturbance. The down-drafting disturbance is usually weak while moving towards the denser lower solar atmosphere, therefore, it is not seen well in form of significant density perturbations below in the density maps. However, its effect can be visible only once it rebounds back at certain later time and helps in the formation of the secondary jets and interaction with the falling material
of the first jet. We study the propagation and evolutionary properties of the first upward propagating isolated jet (at different pressure pulse strength and different magnetic field), therefore, we do not consider other secondary jets and their evolution. Fig. 9 displays an interesting correlation between maximum height of the jets and strength of the pressure pulse, which shows a linear increasing trend. This directly infers that if the extent of the heating and thus pressure perturbations will be higher then more longer jets can be generated from the same location in the chromosphere. For the strong magnetic field, the height of
the jet will be longer at a particular pressure pulse strength compared to the same for the weaker magnetic field strength. The width of the jets may be determined by the conservation of mass in the cross section of the flowing jet spire, however, many other factors, i.e., the configuration of the magnetic field, the interaction of counter-propagating disturbances near the base of the jet etc can affect its value. The width of the jets is measured by using **Gaussian fit to the spatial profile of the jets when they attain maximum height. The example spatial profiles of various jets when they reach at their respective maximum**
**heights are shown in Fig. 10 (top-panel). Since the base of these jets exhibits complex shape and motions, therefore,**

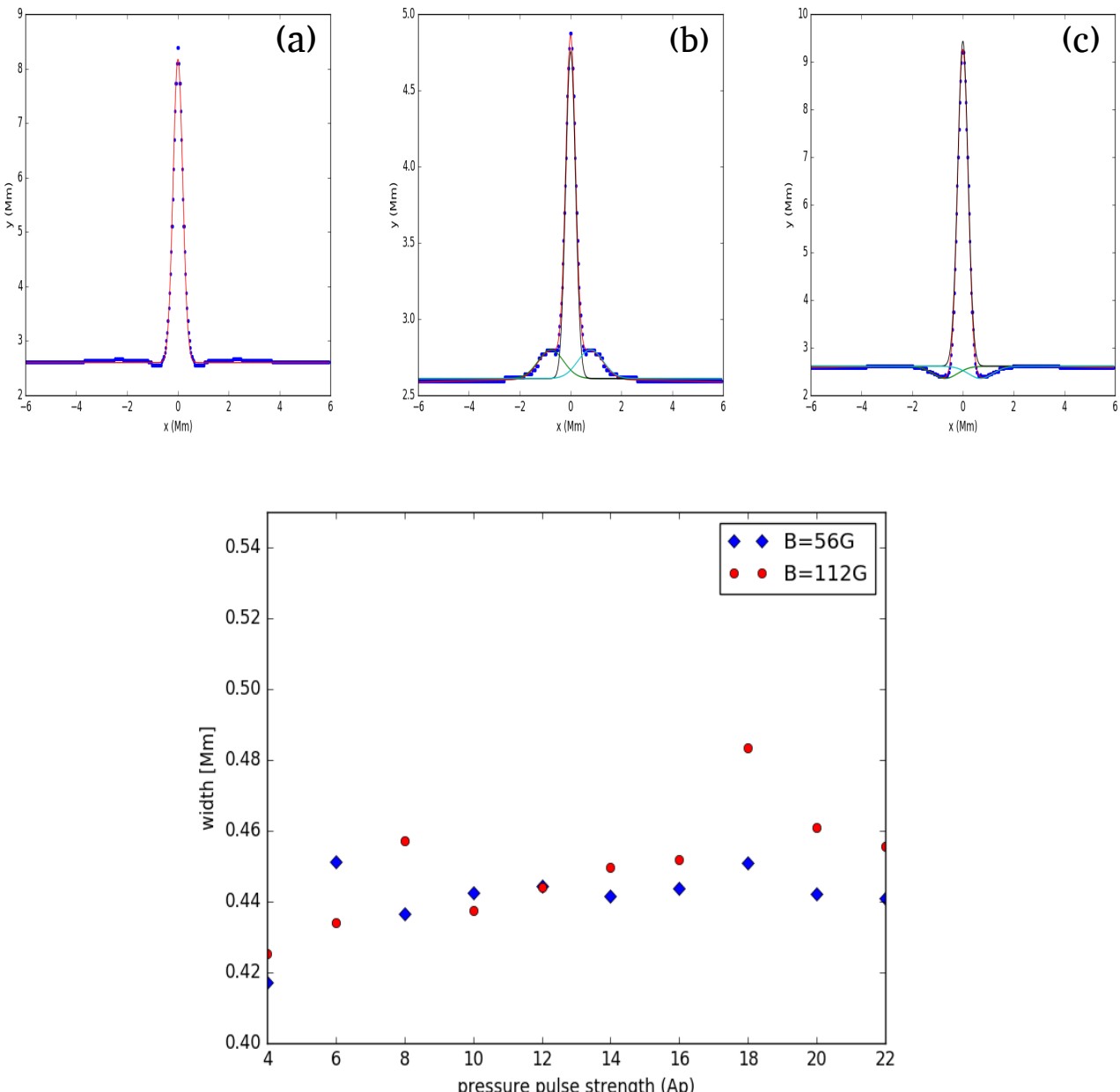

**Figure 10. Top-panel: The example spatial profiles of various jets when they reach at their respective maximum height In the top-panel, figures 'a', 'b', 'c' respectively show spatial profiles of the jet at B=56 Gauss, $A_p$=16; B=56 Gauss, $A_p$=6,B=112 Gauss, $A_p$=18. Since the base of these jets exhibits complex shape, therefore, the triple or single Gaussian profiles are fitted and the respective FWHMs are estimated after determining the Gaussian width.** Bottom-panel: Width of the jets vs. pressure pulse strength (Ap) in the solar atmosphere for magnetic field B= 56 Gauss (blue-diamonds) and B=112 Gauss (red circles). **The FWHM of various jets lie between 0.42 to 0.48 Mm. They show mild increasing trend though for both the magnetic fields.**

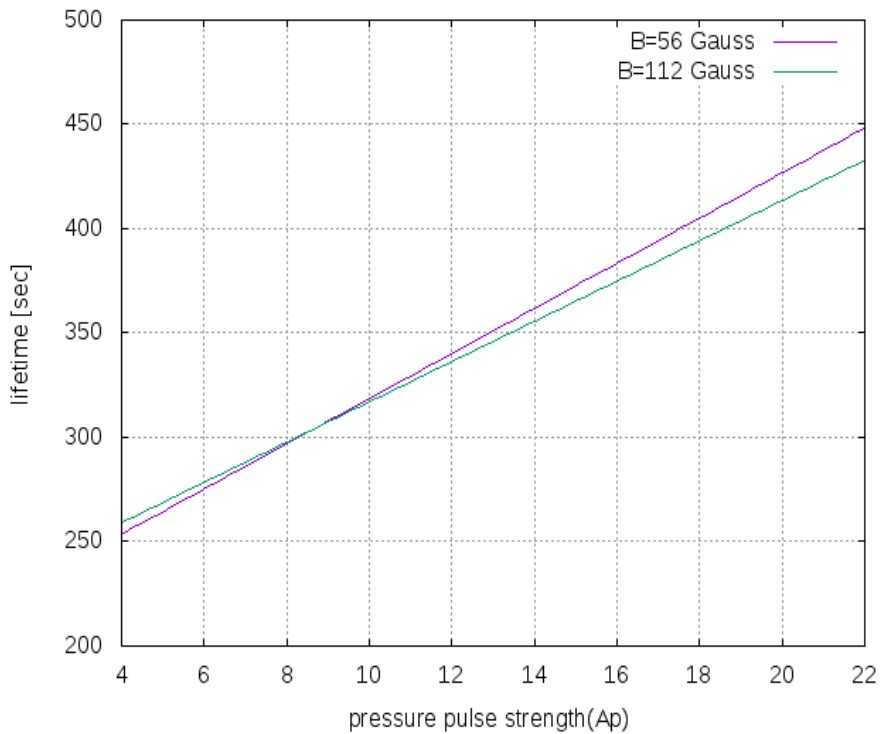

**Figure 11.** Life-time of the model jets vs. pressure pulse strength.

the triple or single Gaussian profiles are fitted on their spatial profiles and the respective FWHMs are estimated after determining the fitted Gaussian width. Width of the jets vs. pressure pulse strength (Ap) in the solar atmosphere for the magnetic field B= 56 Gauss (blue-diamonds) and B=112 Gauss (red circles) are shown in the bottom panel of Fig. 10. The FWHM of various jets lie between 0.42 to 0.48 Mm. Although, they exhibit on average mild increasing trend for both the magnetic fields. The life-time of the jets shows linear variation with a positive slope w.r.t. the pressure pulse strength (Fig. 11). This scenario depicts that more the localized energy release in the chromosphere, occurs the more larger perturbation will lead longer jets with long life-time. Moreover, the lifetime of the jet for a particularly larger pressure pulse strength will be smaller in the presence of strong magnetic field and vice-versa ($A_p \geq 10$). However in our simulation under the given initial condition, the life-time is almost same for the two field strengths, particularly at low amplitude pressure perturbation ($A_p \leq 10$).

## 4   Conclusions

There are several jets in the chromosphere (e.g., macrospicules, network jets, isolated repeated jets, confined surges; (e.g., Wilhelm, 2000; Murawski et al., 2011; Uddin et al., 2012; Kayshap et al., 2013a, 2018; Li et al., 2019, and references cited

therein). The common resemblance point in the evolution of these jets is the presence of heating at their base. Sometimes these jets are so abundant that they over-impose on the co-spatial presence of classical chromospheric ejecta like type-I and type-II spicules (Tian et al., 2014). Therefore, we can not ignore the role of such episodic jets in determining the mass as well as energy transport in the corona. It should be noted that we attempt here to model the above mentioned episodic and impulsive jets in terms of their kinematical and evolutionary properties. However we exclude the classical chromospheric spicules (type-I and type-II), spicule-like structures, anemone jets, etc in this physical model as they may require additional driving mechanisms for their launch and in the non-linear regime complex plasma flows, and wave activity may present there (e.g., De Pontieu et al., 2004; Shetye et al., 2016; Srivastava et al., 2017; Martínez-Sykora et al., 2017, and references cited therein). Therefore, we simply include those jets and their kinematical properties in our model, which evolve due to heating at their footpoints and associated pressure pulse.

It should be noted that our results demonstrate that the elongation and life-time of these jets are directly proportional to the heating pulse, and their shape depends upon the complex plasma motions near their base as well as magnetic field configurations of their spire. Therefore, in general we see the brightening and mass motions starting at their base and evolution of the plasma motion occurs at the Jet's spire. Our model represents the formation of such chromospheric jets which are evolved due to the pressure perturbations near their footpoints. The pressure pulse driven jets are most likely driven by the localized heating at their footpoints lying in the lower solar atmosphere. These perturbations are converted into magneto-acoustic shocks when they move up in the stratified solar atmosphere. The low-pressure region created behind the shock is followed by the motion of cool plasma along the field lines that form the jet. The wave-driven jets are usually launched by the evolution of the magnetic as well as plasma perturbations both. When the velocity, as well as pressure perturbations, are launched higher in the solar atmosphere, they can generate MHD modes in the structured solar atmosphere. In the case of only the Lorentz magnetic force and related velocity perturbations, the $Alfvén$ modes can be generated. These waves under the non-linear effects can also be associated with the ponderomotive force guiding vertical plasma flows in form of jets (e.g., Murwaski et al. 2014, 2018). The strong magnetic field guides more longer jets with a comparatively shorter life-time particularly at large pressure perturbations (Fig. 9, 11). **However, the width of these jets for different pressure pulse strength lie in the range of 0.42 to 0.48 Mm range for strong (e.g., B=112 Gauss) and weaker (e.g., B=56 Gauss) magnetic fields (Fig. 10)**. Although, we present the kinematics and evolutionary properties of the a single isolated jet, but usually such sites where pressure perturbations do occur, exhibit the formation of multiple jets associated with the significant brightening at their base. Such locations are very common sites for the origin of the impulsively generation of isolated chromospheric jets as mentioned above.

*Author contributions.* All authors have contributed in a equal manner.

*Competing interests.* There is no competing interest in the authors.

*Acknowledgements.* We acknowledge the UKIERI Research grant provided by University Grant Commission (UGC), India and British Council, UK for the support of current research. We thankfully acknowledge the support of Dr. P. Konkol and Prof. K. Murawski, UMCS, Lublin, Poland for providing realistic solar atmosphere to incorporate into the PLUTO code. B.S. acknowledge the Human Resource Development Group (HRDG), Council of Scientific & Industrial research (CSIR) for providing him a research scholar grant. B.S. thanks the Department of Physics, Indian Institute of Technology (BHU) for providing him a research facilities. We also thanks both the referees for their constructive comments and editor Dr. S. Shelyeg for his valuable remarks that improved our manuscript. Finally, we acknowledge the use of PLUTO code in our present work.

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
