# Peer review of "On modelling the kinematics and evolutionary properties of pressure pulse driven impulsive solar jets"

_Annales Geophysicae, 2019_

## Referee Comment (RC1) · Anonymous Referee #1 · 23 May 2019

This work deals with the parametric study of the pressure pulse driven solar jets in the chromosphere. The work involves realistic solar atmosphere and magnetic field mimicking the open and gravitationally stratified open field magnetoplasma system of the quiet-Sun atmosphere. The model is detailed study of the kinematics of the cool chromospheric jets which could be generated by the localized heating and the pressure perturbations. This model extends the previous works on the pressure pulsle driven spicules to a more general cases applied for the variety of the solar jets like marcospicules, network jets, cool jets, confined surges etc with the application of realistic solar atmosphere and temperature profile. The implementation of Transition Region also brings the more appropriate results. Paper deals with the kinematical properties

and morphological evolutions of the cool jets that can be applicable to the variety of chromospheric and TR jets. The paper is clearly written and basic results are presented very well. I do not offer any major revision and recommend the paper for its publications. However, authors should explain/implement some minor comments before its final publication.

(1) In the introduction, author should include/discuss some more works that are focused on "pressure pulse driven solar jets", e.g., Martinez-sykora et al. 2011, Judge et al. 2012, Kuzma et al. 2017

(2) Explain why you specifically chosen certain height for pressure pulse?

(3) Why the particular pulse is chosen?

(4) Please describe a little more about the response of chromosphere pressure disturbance.

(5) What is the physical implication of chosen magnetic field?

(6) Describe a little more specifically that how pressure driven jets are more likely in the chromsophere than other wave driven jets.

Apart from these comments, Editor may also suggest the authors to improve the English of the paper at some stage.
* * *

---

## Referee Comment (RC2) · Anonymous Referee #2 · 13 Jun 2019

Referee report manuscript On Modeling kinematic and evolutionary properties of pressure pulse driven impulsive solar jets Balveer Singh, Kushagra Sharma, Abhishek K. Srivastava

This paper presents a numerical modeling of pressure-driven solar jets. More in particular, by using a simplified model of the solar atmosphere with open and expanding field lines, but including the effects of a realistic temperature variation with height, they study the dynamics of pressure-driven pulses mimicking solar jets. The pressure perturbations to the equilibrium atmosphere have a gaussian form and are launched in the chromosphere, mimicking localized heating events. The manuscript provides a series

of interesting dynamic properties of these jets, which can play an important role in the mass deposition and energy transport to the upper layers of the solar atmosphere. In addition, it is found that these jets exhibit asymmetric parabolic paths. While I find the paper timely, clear, coincise and matching the criteria of scientific correctness, I also think that with some small effort it can be improved a lot. I therefore encourage the authors to consider the following suggestions and comments. In addition, I also provide a list of minor corrections.

Major comments: - Fig. 1 shows magnetic field geometry. The evolutionary properties of jets may depend not only on this but also on the strength of the background field, as well as other background physical quantities. I kindly ask the authors to show maps of the background quantities as well, as they can have a direct impact on the results. In this regard, it would be also worthwhile to discuss, and possibly study, the effects of a different background field strengths on the evolution of jets.

- I find odd the adoption of a threshold based on the RGB values of the maps in Fig. 3 to track the jets. Indeed, this is not a physical quantity and color bars in figure 3 are not even shown to help the reader.

- It appears that the data points in Fig. 7 have nothing to do with a linear trend. I therefore recommend not to try to fit any linear function to them.

- Again in fig. 7, there is a big jump between Ap=10 and Ap=12 which I do not physically understand and is not commented in the text.

- Page 9 lines 7-9. The effects of the downward propagating counterpart of the perturbation is mentioned but, there is not supporting plot showing the temporal evolution of the jets and, in particular, this effects.

- It would be nice to show the evolution of the pulses for different quantities (e.g. Bz). This would increase the value of the results and provide a more complete characterization of the process.

[Figure]

Minor comments: - End of Page 4. It is said that the vertical coordinate of magnetic pole is in the convection zone. It would be good to extend the axis range of Fig. 1 to this layer for consistency.

- In order to be consistent with the notation of equations in Sect. 2.1.1, the pressure pulse in Fig. 3 should be indicated as Ap=4–22 not "p"

- The size of the figures is not adequate to the data they contain (i.e. sometimes their are too big)

---

## Author Comment (AC1) · 3 Aug 2019

Dear Editor; Thanks for your email and the referee's comments. Here we reply to all the mentioned comments by the editor.

1st Comment: In the introduction, the author should include/discuss some more works that are focused on "pressure pulse driven solar jets", e.g., Martinez-Sykora et al. 2011, Judge et al. 2012, Kuzma et al. 2017.

Reply:-

Martinez-sykora et al. (2011) and Judge et al. (2012) have studied the structure and

evolution of the spicule-like jets that are triggered by the pulses which are launched in the vertical velocity component from the upper chromosphere. The vertical velocity pulse is converted into a shock that propagates in the upward direction by which chromospheric plasma follows these shocks and comes out into the corona (Martinez-Sykora et al. 2009A, McIntosh et al. 2007 and references cited therein). These shock-driven spicules like jets show a narrow linear structure that reaches a few Mm above the chromosphere. B. Kuźma et al. (2017) also proposed the model for spicules like jets which are triggered by the vertical velocity pulses, while radiative cooling and thermal conduction terms do not exert any significant role in the dynamics of these type of spicules.

We have written these descriptions in the Introduction of the manuscript.

2nd Comment: Explain why you specifically chosen certain height for pressure pulse?

Reply:-

The pressure pulse driven cool jets depend on the amplitude of the initial pulse (Ap) as well as the height of its triggering. This further converts into the acoustic shocks that trigger plasma into the upward direction. The low-pressure region is created behind the slow magnetoacoustic shock, which drives the cool plasma from the chromosphere upward. So the amplitude of the pressure pulse, as well as its triggering site, are the important factors on which the evolution of the jets depends. In this model, we have studied chromospheric cool jets, so we can take a certain amplitude of the pressure pulse at the chromospheric height below the TR to generate these jets.

We have added this description in the revised manuscript by citing relevant references also.

3rd Comment: Why the particular pulse is chosen?

Reply:-

Observations reveal that these type of chromospheric jets are found to be associated

with the impulsive origin in the chromosphere, and their footpoints are usually associated with the brightening. The chosen pressure pulses in this modeling mimic the heating scenario at their footpoints before their evolution. This is not a direct implementation of the heating as we use the ideal set of MHD equations in the numerical simulation. However, the Gaussian pressure pulse implies the disturbance that represents the response in the form of the pressure perturbation once the underlying heating episode is completed. This perturbation locally alters the equilibrium and generates the propagating perturbations. The pressure pulse is chosen with varying amplitude Ap = 4-22 to understand the evolutionary and propagation properties of the cool chromospheric jets.

We have added these justifications in the revised manuscript.

4th Comment: Please describe a little more about the response of chromosphere pressure disturbance.

Reply:-

These perturbations are converted into magneto-acoustic shocks when they move up in the stratified solar atmosphere. The low-pressure region created behind the shock is followed by the motion of cool plasma that forms the jets.

These physics-related arguments are added in the revised manuscript.

5th Comment: What is the physical implication of the chosen magnetic field?

Reply:-

The quiet-Sun chromospheric cool jets are simulated in our model. We set the source magnetic field of the typical order in the quiet-Sun. Moreover, the chosen magnetic fields smoothly extend to the inner corona and appropriately set the reasonable values of plasma beta and Alfven speed. This helps in the evolution of the perturbations and the launch of the collimated jets. In this model, we have chosen two different configurations of magnetic field strength, i.e., B=56 Gauss and B=112 Gauss.

We have added these justifications in the revised manuscript.

6th Comment: Describe a little more specifically that how pressure-driven jets are more likely in the chromosphere than other wave-driven jets.

Reply:-

The pressure pulse driven jets are most likely driven by the localized heating at their footpoints lying in the lower solar atmosphere. These perturbations are converted into magneto-acoustic shocks when they move up in the stratified solar atmosphere. The low-pressure region created behind the shock is followed by the motion of cool plasma along the field lines that form the jet.

The wave-driven jets are usually launched by the evolution of the magnetic as well as plasma perturbations both. When the velocity, as well as pressure perturbations, are launched higher in the solar atmosphere, they can generate MHD modes in the structured solar atmosphere. In the case of only the Lorentz magnetic force and related velocity perturbations, the Alfven modes can be generated. These waves under the non-linear effects can be associated with the ponderomotive force that can guide vertical plasma flows in the form of the jets.

We have added relevant discussion as per the suggestion of the referee and cited the references in the revised manuscript.

Please also note the supplement to this comment:
https://www.ann-geophys-discuss.net/angeo-2019-67/angeo-2019-67-AC1-supplement.pdf

---

## Author Comment (AC2) · 3 Aug 2019

Dear Editor; Thanks for your email and the referee's comments. Here we reply to all the mentioned comments by the editor.

Major Comments:-

1st Comment: Fig. 1 shows magnetic field geometry. The evolutionary properties of jets may depend not only on this but also on the strength of the background field, as well as other background physical quantities. I kindly ask the authors to show maps of the background quantities as well, as they can have a direct impact on the results. In

this regard, it would be also worthwhile to discuss, and possibly study, the effects of different background field strengths on the evolution of jets.

Reply:-

We are agreed with the referee, and even we have made the simulations for the two different magnetic field strength to compare the results.

As per the suggestions of the referee, we have shown now the background physical quantities (density profile, pressure profile, plasma-beta profile, magnetic field, sound speed, Alfven speed) for the model gravitationally-stratified and magnetized solar atmosphere. We have shown these profiles for the atmosphere which have source magnetic field value of 112 Gauss.

(i) Mass density profile:

The mass density at $y = 2.7$ Mm (at the transition region) is $4.595 * 10\text{-}12$ kg m-3. It grows with the depth and attains $1.537*10\text{-}7$ kg m-3 at $y = 1.5$ Mm (which is located within the chromosphere). Figure 1. Equilibrium profiles of mass density vs. height in the solar atmosphere.

(ii) Gas pressure profile:

The gas pressure at $y = 2.7$ Mm (at the transition region) is $1.26 * 10\text{-}3$ Pascal. It grows with the depth and attains $6.411*10\text{-}1$ Pascal at $y = 1.5$ Mm (which is located within the chromosphere). Figure 2. Equilibrium profiles of gas pressure vs. height in the solar atmosphere.

(iii) Magnetic Field profile:

The magnetic field at $y = 2.7$ Mm (at the transition region) is 6.884 Gauss while It grows with the depth and attains 7.646 Gauss at $y = 1.5$ Mm, which is located within the chromosphere. Figure 3. Magnetic field profiles (By) vs. height (y) in the solar atmosphere.

(iv) Plasma-beta profile:

The plasma-beta at y = 2.7 Mm (at the transition region) is 0.0005 while It grow with the depth and attains 0.219 at y = 1.5 Mm, which is located within the chromosphere. Figure 4. Plasma-beta vs. height (y) in the solar atmosphere.

(v) Sound speed profile:

The sound speed at y = 2.7 Mm (Transition region) is 21.392 km-sec-1 while it decreases with the depth and attains 2.638 km sec-1 at y = 1.5 Mm which is located within the chromosphere. Figure 5. Sound speed (Vs) vs. height (y) in the solar atmosphere.

(vi) Alfven speed profile:

The Alfven speed at y = 2.7 Mm (Transition region) is 1015.583 km sec-1 while it decreases with the depth and attains 6.167 km sec-1 at y = 1.5 Mm which is located within the chromosphere. Figure 6. Alfven speed (VA) vs. height (y) in the solar atmosphere.

The various physical quantities, their values, and variations with the height in the structured solar atmosphere clearly indicate their smooth extension into the inner corona. Their reasonable values are set for the model atmosphere and are appropriate for launching the perturbations and associated jets.

2nd Comment: I find odd the adoption of a threshold based on the RGB values of the maps in Fig. 3 to track the jets. Indeed, this is not a physical quantity and color bars in figure 3 are not even shown to help the reader.

Reply:-

Yes, the color of the maps (RGB Values) is not a physical quantity but in the map, we are plotting it for representing the density. Different values of density have been assigned a different RGB value as it is a color map, so indirectly we can say that color

(RGB Values) are related to the density values. Also, we are attaching color bars scheme for the continuous range of density. We attach all the different plots (density maps) of the jets with the color bars. The RGB analyses have already been explained in the manuscript. Figure 7. Color bars scheme (RGB value) of density map.

3rd Comment: It appears that the data points in Fig. 7 have nothing to do with a linear trend. I, therefore, recommend not to try to fit any linear function to them.

Reply:- We show the width of these jets w.r.t. pressure pulse strength(Ap). We have revisited carefully the width estimation after re-running the simulation for two different magnetic field strengths. Kindly see the revised plot in the manuscript as well as below in the reply.

4th Comment: Again in fig. 7, there is a big jump between Ap=10 and Ap=12 which I do not physically understand and is not commented in the text.

Reply:-

As we mentioned above that we have made the simulation of various jets for two different magnetic field strengths. We have re-estimated the width carefully for both the cases. For different pulse strength, the width of the jets varies from 0.8 Mm to 1.0 Mm. This trend is almost the same for both the magnetic field strengths. Earlier the range of the width was also between 0.65 Mm to 0.9 Mm. Therefore, quantitatively our result remains the same. However, it is found that the width of jets w.r.t pressure pulse strength is almost constant for two different configurations of the magnetic field(B= 56 Gauss and also B=112 Gauss). Figure 8. Width of jets vs. pressure pulse strength (Ap) in the solar atmosphere for magnetic field B= 56 Gauss and B=112 Gauss.

5th Comment: Page 9 lines 7-9. The effects of the downward propagating counterpart of the perturbation are mentioned but, there is not supporting plot showing the temporal evolution of the jets and, in particular, these effects.

Reply:-

We are interested in the evolution of the first isolated jet that is generated due to the forward propagating components of the disturbance. The down-drafting disturbance is usually weak while moving towards the denser lower solar atmosphere, therefore, it is not seen well in the form of significant density perturbations below in the density maps. However, its effect can be visible only once it rebounds back at certain time later and will help in the formation of secondary jets coming from below and interacting with the falling material of the first jet. However, since we are studying the propagation and evolutionary properties of the first upward propagating isolated jet (at different pressure pulse strength and different background magnetic field), therefore, we do not consider the other secondary jets and their evolution.

We have added this justification in the revised manuscript also.

6th Comment: It would be nice to show the evolution of the pulses for different quantities (e.g. Bz). This would increase the value of the results and provide a more complete characterization of the process.

Reply:-

We investigate and show the evolution of the pulses and associated jets at two different configurations of the magnetic field (B = 56 Gauss and B = 112 Gauss) and their mosaic diagram. The automatically detected height-time profiles (see Figure 10) for the jets are originated by imposing different pressure pulses. We also show the profiles of the max. Height (Figure 11), lifetime (Figure 12), and their width(Figure 12) w.r.t. pressure pulse strength for these two magnetic fields, and also compare them.

Figure 9. Evolution of the plasma jets at different pressure pulse and two different configurations of magnetic fields B=56 Gauss and B=112 Gauss. The Mosaic diagram shows the maximum height of the evolved jets at different pressure pulses, e.g., Ap=4, Ap=6, Ap=8, Ap=10, Ap=12, Ap=14, Ap=16, Ap=18, Ap=20, and Ap=22. Horizontal (x) and vertical(y) axes are in Mm.

Figure 10. Height of the jets and their evolution w.r.t. the time Figure 11. Maximum height of the model jets vs. pressure pulse strength(Ap). Figure 12. Life-time of the model jets vs. pressure pulse strength (Ap). Figure 13. Width of the jets vs. pressure pulse strength(Ap) in the solar atmosphere for two different configurations of the magnetic field (B= 56 Gauss and B=112 Gauss).

We have put the revised version of these plots in the manuscript now, and explained them physically. Moreover, we compared the variations that we obtained for two background magnetic field strengths, i.e., B=56 Gauss and 112 Gauss.

Minor Comments:-

1st Comment: End of Page 4. It is said that the vertical coordinate of the magnetic pole is in the convection zone. It would be good to extend the axis range of Fig. 1 to this layer for consistency.

Reply:-

Yes, the magnetic pole is set in the convection zone. The jet is triggering still from the chromosphere (1.8 Mm) along with the open magnetic field lines and the pulse is initially lunched there. So, there are no dynamics occurred in the convection zone related to the evolution of these isolated chromospheric jets. Our realistic temperature mode smoothly extends from the convection zone to the photosphere and thereafter it couples to the chromosphere, TR, and corona. In order to save the computing time, we do set lower boundary at the photosphere. For the visualization, we have already given the magnetic field variations in the plot (see above for B=112 Gauss pole strength) starting from the photosphere to the corona. It clearly shows that the magnetic pole is set somewhere in the convection zone deeper, and the magnetic field is smoothly decayed and extended into the solar atmosphere at higher heights exponentially. Putting the pole deep below in the convection zone is the requirement as we keep the magnetic singularity away from the simulation box in order to set an appropriate initial force and current free magnetic fields.

[Figure]

We have added these justifications in the revised manuscript. We request referee to kindly consider our justification and explanations regarding the simulation.

2nd Comment: In order to be consistent with the notation of equations in Sect. 2.1.1, the pressure pulse in Fig. 3 should be indicated as Ap=4–22 not "p".

Reply:- we have now consistent with the notation of equations and pressure pulse strength 'p' replaced by 'Ap' at all positions.

Please also note the supplement to this comment:
https://www.ann-geophys-discuss.net/angeo-2019-67/angeo-2019-67-AC2-supplement.pdf

[Figure]

Fig. 1. Equilibrium profiles of mass density vs. height in the solar atmosphere.

**Fig. 2.** Equilibrium profiles of gas pressure vs. height in the solar atmosphere.

**Fig. 3.** Magnetic field profiles (By) vs. height (y) in the solar atmosphere

**Fig. 4.** Plasma-beta vs. height (y) in the solar atmosphere

[Figure]

**Fig. 5.** Sound speed (Vs) vs. height (y) in the solar atmosphere.

[Figure]

Fig. 6. Alfven speed (VA) vs. height (y) in the solar atmosphere.

**Fig. 7.** Color bars scheme (RGB value) of density map.

**Fig. 8.** Width of jets vs. pressure pulse strength (Ap) in the solar atmosphere for magnetic field B= 56 Gauss and B=112 Gauss.

| B=56G Ap=4 | B=56G Ap=6 | B=56G Ap=8 | B=56G Ap=10 | B=56G Ap=12 | B=56G Ap=14 | B=56G Ap=16 | B=56G Ap=18 | B=56G Ap=20 | B=56G Ap=22 |
|---|---|---|---|---|---|---|---|---|---|
| B=112G Ap=4 | B=112G Ap=6 | B=112G Ap=8 | B=112G Ap=10 | B=112G Ap=12 | B=112G Ap=14 | B=112G Ap=16 | B=112G Ap=18 | B=112G Ap=20 | B=112G Ap=22 |

**Fig. 9.** Evolution of the plasma jets at different pressure pulse and two different configurations of magnetc fields B=56 Gauss and B=112 Gauss. The Mosaic diagram shows the maximum height of the evolved jets

[Figure]

**Fig. 10.** Height of the jets and their evolution w.r.t. the time.

[Figure]

**Fig. 11.** Maximum height of the model jets vs. pressure pulse strength(Ap).

Interactive
comment

**Fig. 12.** Life-time of the model jets vs. pressure pulse strength (Ap)
**Fig. 13.** Width of the jets vs. pressure pulse strength(Ap) in the solar atmosphere for two different configuration of magnetic field (B= 56 Gauss and B=112 Gauss).

**Supplement:**

**On modelling the kinematics and evolutionary properties of pressure pulse driven impulsive solar jets, MS No.: angeo-2019-67**

Balveer Singh, Kushagra Sharma and Abhishek K. Srivastava
Department of Physics, Indian Institute of Technology (BHU), Varanasi-221005, India

**Dear Editor;**
Thanks for your email and referee's comments. Here we reply to all the mentioned comments by editor.

**Major Comments:-**

**1ˢᵗ Comment:**
Fig. 1 shows magnetic field geometry. The evolutionary properties of jets may depend not only on this but also on the strength of the background field, as well as other background physical quantities. I kindly ask the authors to show maps of the background quantities as well, as they can have a direct impact on the results. In this regard, it would be also worthwhile to discuss, and possibly study, the effects of a different background field strengths on the evolution of jets.

**Reply:-**

**We are agreed with the referee, and even we have made the simulations for the two different magnetic field strength to compare the results.**

**As per the suggestions of the referee, we have shown now the background physical quantities (density profile, pressure profile, plasma-beta profile, magnetic field , sound speed, Alfven speed) for the model gravitationally-stratified and magnetized solar atmosphere. We have shown these profiles for the atmosphere which have source magnetic field value of 112 Gauss.**

**(i) Mass density profile:**

**The mass density at y = 2.7 Mm (at the transition region) is $4.595 * 10^{-12}$ kg m$^{-3}$. It grow with the depth and attain $1.537*10^{-7}$ kg m$^{-3}$ at y = 1.5 Mm (which is located within the chromosphere).**

[Figure]

**Figure 1. Equilibrium profiles of mass density vs. height in the solar atmosphere.**

**(ii) Gas pressure profile:**

**The gas pressure at y = 2.7 Mm (at the transition region) is 1.26 * $10^{-3}$ Pascal. It grow with the depth and attan 6.411*$10^{-1}$ Pascal at y = 1.5 Mm (which is located within the chromosphere).**

[Figure]

**Figure 2. Equilibrium profiles of gas pressure vs. height in the solar atmosphere.**

**(iii) Magnetic Field profile:**

The magnetic field at y = 2.7 Mm (at the transition region) is 6.884 Gauss while It grow with the depth and attains 7.646 Gauss at y = 1.5 Mm, which is located within the chromosphere .

[Figure]

**Figure 3. Magnetic field profiles (By) vs. height (y) in the solar atmosphere.**

**(iv) Plasma-beta profile:**

The plasma-beta at y = 2.7 Mm (at the transition region) is 0.0005 while It grow with the depth and attains 0.219 at y = 1.5 Mm, which is located within the chromosphere .

[Figure]

**Figure 4. Plasma-beta vs. height (y) in the solar atmosphere.**

**(v) Sound speed profile:**

The sound speed at y = 2.7 Mm (Transition region) is 21.392 km-sec$^{-1}$ while it decrease with the depth and attain 2.638 km sec$^{-1}$ at y = 1.5 Mm which is located within the chromosphere.

[Figure]

**Figure 5. Sound speed (V$_s$) vs. height (y) in the solar atmosphere.**

**(vi) Alfven speed profile:**

The Alfven speed at y = 2.7 Mm (Transition region) is 1015.583 km sec$^{-1}$ while it decrease with the depth and attain 6.167 km sec$^{-1}$ at y = 1.5 Mm which is located within the chromosphere.

[Figure]

**Figure 6. Alfven speed (V$_A$) vs. height (y) in the solar atmosphere.**

**The various physical quantities, their values, and variations with the height in the structured solar atmosphere clearly indicate their smooth extension into the inner corona. Their resonable values are set for the model atmosphere, and are appropriate for launching the perturbations and associated jets.**

**2nd Comment:**

I find odd the adoption of a threshold based on the RGB values of the maps in Fig. 3 to track the jets. Indeed, this is not a physical quantity and color bars in figure 3 are not even shown to help the reader.

**Reply:-**

**Yes, color of the maps (RGB Values) are not a physical quantity but in the map we are plotting it for representing the density. Different values of density have been assigned a different RGB value as it is a color map, so indirectly we can say that color (RGB Values) are related to the density values. Also, we are attaching color bars scheme for the continuous range of density.**
**We attach all the different plots (density maps) of the jets with the color bars. The RGB analyses has already been explained in the manuscript.**

[Figure]

**Figure 7. Color bars scheme (RGB value) of density map.**

**3rd Comment:**

It appears that the data points in Fig. 7 have nothing to do with a linear trend. I therefore recommend not to try to fit any linear function to them.

**Reply:-**

**We show the width of these jets w.r.t. pressure pulse strength(Ap). We have revisited carefully the width estimation after re-running the simulation for two different magnetic field strengths. Kindly see the revised plot in the manuscript as well as below in the reply.**

**4th Comment:** Again in fig. 7, there is a big jump between Ap=10 and Ap=12 which I do not physically understand and is not commented in the text.

**Reply:-**

**As we mentioned above that we have made simulation of various jets for two different magnetic field strenghts. We have re-estimated the width carefully for both the cases. For different pulse strengths, the width of the jets varies from 0.8 Mm to 1.0 Mm. This trend is almost same for both the magnetic field strenghts. Earlier the range of the width was also between 0.65 Mm to 0.9 Mm. Therefore, quantitatively our result remains same. However, it is found that the width of jets w.r.t pressure pulse strength is almost constant for two different configuration of magnetic field(B= 56 Gauss and also B=112 Gauss).**

[Figure]

**Figure 8. Width of jets vs. pressure pulse strength (Ap) in the solar atmosphere for magnetic field B= 56 Gauss and B=112 Gauss.**

**5th Comment:**

Page 9 lines 7-9. The effects of the downward propagating counterpart of the perturbation is mentioned but, there is not supporting plot showing the temporal evolution of the jets and, in particular, this effects.

**Reply:-**

**We are interested in the evolution of the first isolated jet that is generated due to the forward propagating components of the disturbance. The downdrafting disturbance is usually weak while moving towards the denser lower solar atmosphere, therefore, it is not seen well in form of significant density perturbations below in the density maps. However, its effect can be visible only once it rebounds back at certain time later and will help in the formation of secondary jets coming from below and interacting with the falling material of the first jet. However, since we are studying the propagation and evolutionary properties of the first upward propagating isolated jet (at different pressure pulse strength and different background magnetic field), therefore, we do not consider the other secondary jets and their evolution.**

**We have added this justification in the revised manuscript also.**

**6th Comment:** It would be nice to show the evolution of the pulses for different quantities (e.g. Bz). This would increase the value of the results and provide a more complete characterization of the process.

**Reply:-**

**We investigate and show the evolution of the pulses and associated jets at two different configurations of the magnetic field (B = 56 Gauss and B = 112 Gauss) and their mosaic diagram. The automatically detected height-time profiles for the jets are originated by imposing different pressure pulses. We also show the profiles of max. height, lifetime, and their width w.r.t. pressure pulse strength for these two magnetic fields, and also compare them.**

[Figure]

**Figure 9. Evolution of the plasma jets at different pressure pulse and two different configurations of magnetc fields B=56 Gauss and B=112 Gauss. The Mosaic diagram shows the maximum height of the evolved jets at different pressure pulses, e.g., Ap=4, Ap=6, Ap=8, Ap=10, Ap=12, Ap=14, Ap=16, Ap=18, Ap=20 and Ap=22. Horizontal (x) and vertical(y) axes are in Mm.**

[Figure]

**Figure 10. Height of the jets and their evolution w.r.t. the time**

[Figure]

**Figure 11. Maximum height of the model jets vs. pressure pulse strength(Ap).**

[Figure]

**Figure 12. Life-time of the model jets vs. pressure pulse strength (Ap).**

[Figure]

**Figure 13. Width of the jets vs. pressure pulse strength(Ap) in the solar atmosphere for two different configuration of magnetic field (B= 56 Gauss and B=112 Gauss).**

**We have put the revised version of these plots in the manuscript now, and explained them physically. Moreover, we compared the variations that we obtained for two background magnetic field strenghts, i.e., B=56 Gauss and 112 Gauss.**

**Minor Comments:-**

**1$^{st}$ Comment:**
End of Page 4. It is said that the vertical coordinate of magnetic pole is in the convection zone. It would be good to extend the axis range of Fig. 1 to this layer for consistency.

**Reply:-**

**Yes, the magnetic pole is set in the convection zone. The jet is triggering still from the chromosphere (1.8 Mm) along with the open magnetic field lines and the pulse is initially lunched there. So, there is no dynamics occurred in the convection zone related to the evolution of these isolated chromospheric jets. Our realistic temperature mode smoothly extends from the convection zone to the photosphere and thereafter it couples to the chromosphere, TR,**

and corona. In order to save the computing time we do set lower boundary at the photosphere. For the visualiztion we have already given the magnetic field variations in the plot (see above for B=112 Gauss pole strength) starting from the phtosphere to the corona. It clearly shows that magnetic pole is set somewhere in the convection zone deeper, and the magnetic field is smoothly decayed and extended into the solar atmosphere at higher heights exponetially. Putting the pole deep below in the convection zone is the requirement as we keep the magnetic sigularity away from the simulation box in order to set an appropriate initial force and current free magnetic fields.

We have added these justification in the revised manuscript. We request referee to kindly consider our justification and explanations regarding the simulation.

**2$^{nd}$ Comment:** In order to be consistent with the notation of equations in Sect. 2.1.1, the pressure pulse in Fig. 3 should be indicated as Ap=4–22 not "p".

**Reply:-**
we have now consistent with the notation of equations and pressure pulse strength 'p' replaced by '$A_p$' at all positions.

---

## Author Response (AR1)

**On modelling the kinematics and evolutionary properties of pressure pulse driven impulsive solar jets, MS No.: angeo-2019-67**

Balveer Singh, Kushagra Sharma and Abhishek K. Srivastava
Department of Physics, Indian Institute of Technology (BHU), Varanasi-221005, India

Dear Editor;
Thanks for your email and constructive comments. Here we reply to all the mentioned comments. We have made the suggestes corrections in the revised manuscript.

**1$^{st}$ Comment:**
Please correct the MHD system (page 3, line 1). There is the whole term missing; also please make a clear difference between vectors and scalars throughout the manuscript. Please also check the text around the MHD system - there are problems, for example I (unit matrix) and p_t are not used, while identified in the text. This is due to aforementioned mistake. Note that it should be p_t instead of pt.

**Reply:**

$$\frac{\partial}{\partial t}\begin{pmatrix}\rho\\\rho\mathbf{v}\\E\\\mathbf{B}\end{pmatrix}+\nabla\cdot\begin{pmatrix}\rho\mathbf{v}\\\rho\mathbf{vv}-\frac{\mathbf{BB}}{\mu}+\mathbf{I}p_t\\(E+p_t)\mathbf{v}-\frac{\mathbf{B}}{\mu}(\mathbf{v.B})\\\mathbf{vB}\text{-}\mathbf{Bv}\end{pmatrix}=\begin{pmatrix}0\\\rho\mathbf{g}\\\rho\mathbf{v.g}\\0\end{pmatrix}$$

and

$$E=\frac{p}{\gamma-1}+\frac{\rho\mathbf{v}^2}{2}+\frac{\mathbf{B}^2}{2\mu}$$

**Yes, by the mistake, term Ipt is missed. So now, we have added this term in the set of MHD equations and in the revised manuscript. We have shown difference between vectors (bold) and scalars (normal) throughout the manuscript and we have replaced standard natation p_t in place of pt in the revised manuscript.**

**2$^{nd}$ Comment:**
Page 3, line 10: "we do not consider the microscopic interactions"... The ideal equation of state is due to elastic collisions between particles, so assuming an ideal gas law you already assume interaction between particles.

**Reply:**
**we have removed the statement "we do not consider the microscopic interactions of the particles at considered length scale" in the revised manuscript.**

**3rd Comment:**
Page 3 line 21 - "attributed by the measurement of Avrett..." - this temperature profile was not measured, but inferred from the observed line profiles.

**Reply:**

**we have correct and replace this statement by "inferred from the observed line profiles by Avrett and Loeser (2008) and also depicted in Konkol et al. (2012).**
**We have also added this correction in the revised manuscript.**

**4th Comment:**
Page 5 line 6 - there is "s-2", while it should be s^{-2}.

**Reply:**

**we have correct the suggest notation " s–2" in the revised manuscript.**

**5th Comment:**
Page 6 Line 2 - the code cannot be "finite volume and finite difference". It can be "finite volume or finite difference", if it is possible to switch that.

**Reply:**

**Yes, a place of finite volume and finite difference, it should be finite volume / finite-difference.**
**We have corrected this statement in the revised manuscript.**

**6th Comment:**
Page 7 lines 25-30: I still suggest making it properly. I cannot see where in the plot G=255 or 245, because there is no colour bar in this figure. I understand what you are doing there, but ultimately the values, which are shown in the figure, are the values of density. It is the jump of density in the photosphere you see as change in the colour. You do not have to refer to the values of colour to establish the boundary of the simulated jet, you just need to find a threshold value for the density.

**Reply:**

**We have analyzed the same by image processing using Matlab which can also be done using the threshold values for density as suggested by you. Also we have attached the RGB values in color bar alongside the density values for reference in fig. 7.**

[Figure]

**Figure 7. Automated detection of the plasma jet in the numerical simulation data to establish its time-distance profile and termination point.**

**7th Comment:**

Page 11 fig. 6 - I do not understand this figure. How do you measure the width of jet at their maximum height, and what is this exactly? Is this width measured when the jet is highest? Then, at which height the width is measured? Or this width is measured at the tip of the jet? Then it is not a reliable measure. If the former, I suggest adding description on where exactly the width is measured. If the latter, then I would suggest to measure the width at half-maximum instead.

**Reply:**
**The width of the jets is measured by using Gaussian fit to the spatial profile of the jets when they attain maximum height. The example spatial profiles of various jets when they reach at their respective maximum heights are shown in Fig. 10 (top-panel). Since the base of these jets exhibits complex shape and motions, therefore, the triple or single Gaussian profiles are fitted on their spatial profiles and the respective FWHMs are estimated after determining the fitted Gaussian width. Width of the jets vs. pressure pulse strength ($A_p$) in**

the solar atmosphere for the magnetic field B= 56 Gauss (blue-diamonds) and B=112 Gauss (red circles) are shown in the bottom panel of Fig. 10.

The FWHM of various jets lie between 0.42 to 0.48 Mm. Although, they exhibit on average mild increasing trend for both the magnetic fields.

[Figure]

Figure 10. Top-panel: The example spatial profiles of various jets when they reach at their respective maximum height In the top-panel, figures 'a', 'b', 'c' respectively show spatial profiles of the jet at B=56 Gauss, $A_p$=16; B=56 Gauss, $A_p$=6, B=112 Gauss, $A_p$=18. Since the base of these jets exhibits complex shape, therefore, the triple or single Gaussian profiles are fitted and the respective FWHMs are estimated after determining the Gaussian width. Bottom-panel: Width of the jets vs. pressure pulse strength ($A_p$) in the solar atmosphere for magnetic field B= 56 Gauss (blue-diamonds) and B=112 Gauss (red circles). The FWHM of various jets lie between 0.42 to 0.48 Mm. They show mild increasing trend though for both the magnetic fields.

---

## Author Response (AR2)

Dear Editor,
We are very greatfull that you have accepted the manuscript in Annales Giophysicae.

with best regards

Balveer Singh and all Co-Authors.